# DreamGaussian: Generative Gaussian Splatting for Efficient 3D Content Creation

**Jiaxiang Tang[1]**[*], **Jiawei Ren[2], Hang Zhou[3], Ziwei Liu[2], Gang Zeng[1]**
[1]National Key Laboratory of General AI, School of IST, Peking University
[2]S-Lab, Nanyang Technological University     [3]Baidu Inc.

https://dreamgaussian.github.io/

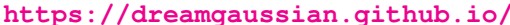
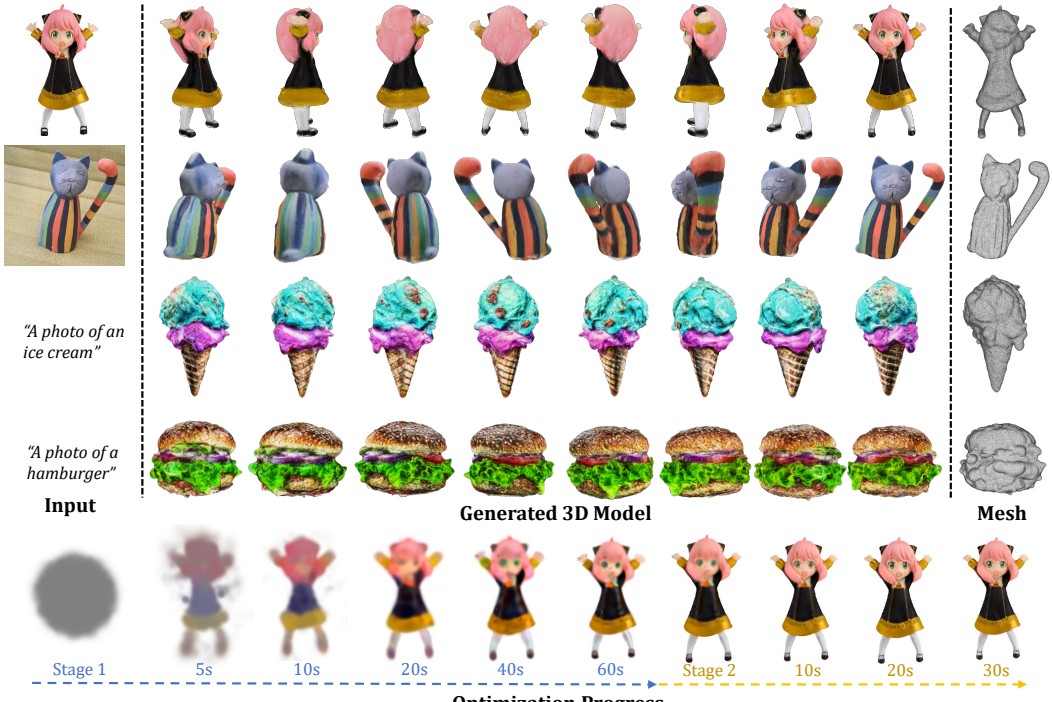

Figure 1: **DreamGaussian** aims at accelerating the optimization process of both image- and text-to-3D tasks. We are able to generate a high quality textured mesh in several minutes.

## Abstract

Recent advances in 3D content creation mostly leverage optimization-based 3D generation via score distillation sampling (SDS). Though promising results have been exhibited, these methods often suffer from slow per-sample optimization, limiting their practical usage. In this paper, we propose **DreamGaussian**, a novel 3D content generation framework that achieves both efficiency and quality simultaneously. Our key insight is to design a generative 3D Gaussian Splatting model with companioned mesh extraction and texture refinement in UV space. In contrast to the occupancy pruning used in Neural Radiance Fields, we demonstrate that the progressive densification of 3D Gaussians converges significantly faster for 3D generative tasks. To further enhance the texture quality and facilitate downstream applications, we introduce an efficient algorithm to convert 3D Gaussians into textured meshes and apply a fine-tuning stage to refine the details. Extensive experiments demonstrate the superior efficiency and competitive generation quality of our proposed approach. Notably, DreamGaussian produces high-quality textured meshes in just 2 minutes from a single-view image, achieving approximately 10 times acceleration compared to existing methods.

---

[*]This work was partly done when interning with Baidu Inc. and visiting NTU S-Lab.

# 1 INTRODUCTION

Automatic 3D digital content creation finds applications across various domains, including digital games, advertising, films, and the MetaVerse. The core techniques, including image-to-3D and text-to-3D, offer substantial advantages by significantly reducing the need for manual labor among professional artists and empowering non-professional users to engage in 3D asset creation. Drawing inspiration from recent breakthroughs in 2D content generation (Rombach et al., 2022), the field of 3D content creation has experienced rapid advancements. Recent studies in 3D creation can be classified into two principal categories: *inference-only 3D native methods* and *optimization-based 2D lifting methods*. Theoretically, 3D native methods (Jun & Nichol, 2023; Nichol et al., 2022; Gupta et al., 2023) exhibit the potential to generate 3D-consistent assets within seconds, albeit at the cost of requiring extensive training on large-scale 3D datasets. The creation of such datasets demand substantial human effort, and even with these efforts, they continue to grapple with issues related to limited diversity and realism (Deitke et al., 2023b;a; Wu et al., 2023).

On the other hand, Dreamfusion (Poole et al., 2022) proposes Score Distillation Sampling (SDS) to address the 3D data limitation by distilling 3D geometry and appearance from powerful 2D diffusion models (Saharia et al., 2022), which inspires the development of recent *2D lifting* methods (Lin et al., 2023; Wang et al., 2023b; Chen et al., 2023c). In order to cope with the inconsistency and ambiguity caused by the SDS supervision, Neural Radiance Fields (NeRF) (Mildenhall et al., 2020) are usually adopted for their capability in modeling rich 3D information. Although the generation quality has been increasingly improved, these approaches are notorious for hours-long optimization time due to the costly NeRF rendering, which restricts them from being deployed to real-world applications at scale. We argue that the occupancy pruning technique used to accelerate NeRF (Müller et al., 2022; Sara Fridovich-Keil and Alex Yu et al., 2022) is ineffective in generative settings when supervised by the ambiguous SDS loss as opposed to reconstruction settings.

In this work, we introduce the *DreamGaussian* framework, which greatly improves the 3D content generation efficiency by refining the design choices in an optimization-based pipeline. Photo-realistic 3D assets with explicit mesh and texture maps can be generated from a single-view image within only 2 minutes using our method. Our core design is to *adapt 3D Gaussian Splatting (Kerbl et al., 2023) into the generative setting with companioned meshes extraction and texture refinement.* Compared to previous methods with the NeRF representation, which find difficulties in effectively pruning empty space, our *generative Gaussian splatting* significantly simplifies the optimization landscape. Specifically, we demonstrate the progressive densification of Gaussian splatting, which is in accordance with the optimization progress of generative settings, greatly improves the generation efficiency. As illustrated in Figure 1, our image-to-3D pipeline swiftly produces a coarse shape within seconds and converges efficiently in around 500 steps on a single GPU.

Due to the ambiguity in SDS supervision and spatial densification, the directly generated results from 3D Gaussians tend to be blurry. To address the issue, we identify that the texture needs to be refined explicitly, which requires delicate textured polygonal mesh extraction from the generated 3D Gaussians. While this task has not been explored before, we design an efficient algorithm for mesh extraction from 3D Gaussians by local density querying. Then a generative UV-space refinement stage is proposed to enhance the texture details. Given the observation that directly applying the latent space SDS loss as in the first stage results in over-saturated blocky artifacts on the UV map, we take the inspiration from diffusion-based image editing methods (Meng et al., 2021) and perform image space supervision. Compared to previous texture refinement approaches, our refinement stage achieves better fidelity while keeping high efficiency.

In summary, our contributions are:

1. We adapt 3D Gaussian splatting into generative settings for 3D content creation, significantly reducing the generation time of optimization-based 2D lifting methods.

2. We design an efficient mesh extraction algorithm from 3D Gaussians and a UV-space texture refinement stage to further enhance the generation quality.

3. Extensive experiments on both Image-to-3D and Text-to-3D tasks demonstrate that our method effectively balances optimization time and generation fidelity, unlocking new possibilities for real-world deployment of 3D content generation.

## 2 RELATED WORK

### 2.1 3D REPRESENTATIONS

Various 3D representations have been proposed for different 3D tasks. Neural Radiance Fields (NeRF) (Mildenhall et al., 2020) employs a volumetric rendering and has been popular for enabling 3D optimization with only 2D supervision. Although NeRF has become widely used in both 3D reconstruction (Barron et al., 2022; Li et al., 2023d; Chen et al., 2022; Hedman et al., 2021) and generation (Poole et al., 2022; Lin et al., 2023; Chan et al., 2022), optimizing NeRF can be time-consuming. Various attempts have been made to accelerate the training of NeRF (Müller et al., 2022; Sara Fridovich-Keil and Alex Yu et al., 2022), but these works mostly focus on the reconstruction setting. The common technique of spatial pruning fails to accelerate the generation setting. Recently, 3D Gaussian splatting (Kerbl et al., 2023) has been proposed as an alternative 3D representation to NeRF, which has demonstrated impressive quality and speed in 3D reconstruction (Luiten et al., 2023). The efficient differentiable rendering implementation and model design enables fast training without relying on spatial pruning. In this work, we for the first time adapt 3D Gaussian splatting into generation tasks to unlock the potential of optimization-based methods.

### 2.2 TEXT-TO-3D GENERATION

Text-to-3D generation aims at generating 3D assets from a text prompt. Recently, data-driven 2D diffusion models have achieved notable success in text-to-image generation (Ho et al., 2020; Rombach et al., 2022; Saharia et al., 2022). However, transferring it to 3D generation is non-trivial due to the challenge of curating large-scale 3D datasets. Existing 3D native diffusion models usually work on a single object category and suffer from limited diversity (Jun & Nichol, 2023; Nichol et al., 2022; Gupta et al., 2023; Lorraine et al., 2023; Zhang et al., 2023; Zheng et al., 2023; Ntavelis et al., 2023; Chen et al., 2023b; Cheng et al., 2023; Gao et al., 2022). To achieve open-vocabulary 3D generation, several methods propose to lift 2D image models for 3D generation (Jain et al., 2022; Poole et al., 2022; Wang et al., 2023a; Mohammad Khalid et al., 2022; Michel et al., 2022). Such 2D lifting methods optimize a 3D representation to achieve a high likelihood in pretrained 2D diffusion models when rendered from different viewpoints, such that both 3D consistency and realisticity can be ensured. Following works continue to enhance various aspects such as generation fidelity and training stability (Lin et al., 2023; Tsalicoglou et al., 2023; Zhu & Zhuang, 2023; Yu et al., 2023; Li et al., 2023c; Chen et al., 2023d; Wang et al., 2023b; Huang et al., 2023; Metzer et al., 2022; Chen et al., 2023c), and explore further applications (Zhuang et al., 2023; Singer et al., 2023; Raj et al., 2023). However, these optimization-based 2D lifting approaches usually suffer from long per-case optimization time. Particularly, employing NeRF as the 3D representation leads to expensive computations during both forward and backward. In this work, we choose 3D Gaussians as the differentiable 3D representation and empirically show that it has a simpler optimization landscape.

### 2.3 IMAGE-TO-3D GENERATION

Image-to-3D generation targets generating 3D assets from a reference image. The problem can also be formulated as single-view 3D reconstruction (Yu et al., 2021; Trevithick & Yang, 2021; Duggal & Pathak, 2022), but such reconstruction settings usually produce blurry results due to the lack of uncertainty modeling. Text-to-3D methods can also be adapted for image-to-3D generation (Xu et al., 2023a; Tang et al., 2023b; Melas-Kyriazi et al., 2023) using image captioning models (Li et al., 2022; 2023a). Recently, Zero-1-to-3 (Liu et al., 2023b) explicitly models the camera transformation into 2D diffusion models and enable zero-shot image-conditioned novel view synthesis. It achieves high 3D generation quality when combined with SDS, but still suffers from long optimization time (Tang, 2022; Qian et al., 2023). One-2-3-45 (Liu et al., 2023a) trains a multi-view reconstruction model for acceleration at the cost of the generation quality. With an efficiency-optimized framework, our work shortens the image-to-3D optimization time to 2 minutes with little sacrifice on quality.

## 3 OUR APPROACH

In this section, we introduce our two-stage framework for efficient 3D content generation for both Image-to-3D and Text-to-3D tasks as illustrated in Figure 2. Firstly, we adapt 3D Gaussian splat-

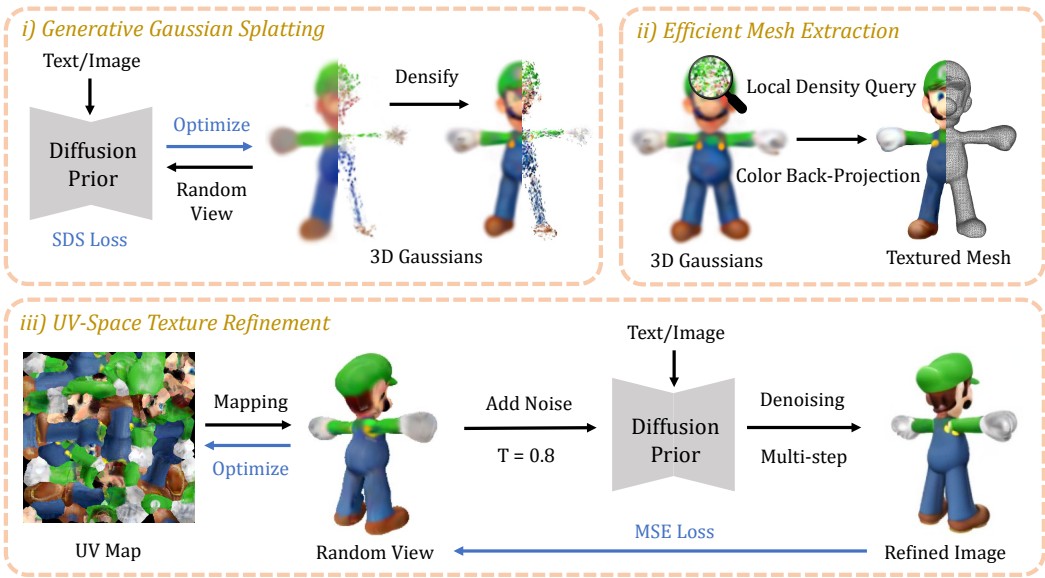

Figure 2: **DreamGaussian Framework**. 3D Gaussians are used for efficient initialization of geometry and appearance using single-step SDS loss. We then extract a textured mesh and refine the texture image with a multi-step MSE loss.

ting (Kerbl et al., 2023) into generation tasks for efficient initialization through SDS (Poole et al., 2022) (Section 3.1). Next, we propose an algorithm to extract a textured mesh from 3D Gaussians (Section 3.2). This texture is then fine-tuned by differentiable rendering (Laine et al., 2020) through a UV-space refinement stage (Section 3.3) for final exportation.

## 3.1 GENERATIVE GAUSSIAN SPLATTING

Gaussian splatting (Kerbl et al., 2023) represents 3D information with a set of 3D Gaussians. It has been proven effective in reconstruction settings (Kerbl et al., 2023; Luiten et al., 2023) with high inference speed and reconstruction quality under similar modeling time with NeRF. However, its usage in a generative manner has not been explored. We identify that the 3D Gaussians can be efficient for 3D generation tasks too.

Specifically, the location of each Gaussian can be described with a center $\mathbf{x} \in \mathbb{R}^3$, a scaling factor $\mathbf{s} \in \mathbb{R}^3$, and a rotation quaternion $\mathbf{q} \in \mathbb{R}^4$. We also store an opacity value $\alpha \in \mathbb{R}$ and a color feature $\mathbf{c} \in \mathbb{R}^3$ for volumetric rendering. Spherical harmonics are disabled since we only want to model simple diffuse color. All the above optimizable parameters is presented by $\Theta$, where $\Theta_i = \{\mathbf{x}_i, \mathbf{s}_i, \mathbf{q}_i, \alpha_i, \mathbf{c}_i\}$ is the parameter for the $i$-th Gaussian. To render a set of 3D Gaussians, we need to project them onto the image plane as 2D Gaussians. Volumetric rendering is then performed for each pixel in front-to-back depth order to evaluate the final color and alpha. In this work, we use the highly optimized renderer implementation from Kerbl et al. (2023) to optimize $\Theta$.

We initialize the 3D Gaussians with random positions sampled inside a sphere, with unit scaling and no rotation. These 3D Gaussians are periodically densified during optimization. Different from the reconstruction pipeline, we start from fewer Gaussians but densify it more frequently to align with the generation progress. We follow the recommended practices from previous works (Poole et al., 2022; Huang et al., 2023; Lin et al., 2023) and use SDS to optimize the 3D Gaussians (Please refer to Section A.1 for more details on SDS loss). At each step, we sample a random camera pose $p$ orbiting the object center, and render the RGB image $I_{\text{RGB}}^p$ and transparency $I_{\text{A}}^p$ of the current view. Similar to Dreamtime (Huang et al., 2023), we decrease the timestep $t$ linearly during training, which is used to weight the random noise $\epsilon$ added to the rendered RGB image. Then, different 2D diffusion priors $\phi$ can be used to optimize the underlying 3D Gaussians through SDS.

**Image-to-3D.** For the image-to-3D task, an image $\tilde{I}_{\text{RGB}}^r$ and a foreground mask $\tilde{I}_{\text{A}}^r$ are given as input. Zero-1-to-3 XL (Liu et al., 2023b; Deitke et al., 2023b) is adopted as the 2D diffusion prior. The

SDS loss can be formulated as:

$$\nabla_\Theta \mathcal{L}_{\text{SDS}} = \mathbb{E}_{t,p,\epsilon} \left[ w(t)(\epsilon_\phi(I^p_{\text{RGB}}; t, \tilde{I}^r_{\text{RGB}}, \Delta p) - \epsilon) \frac{\partial I^p_{\text{RGB}}}{\partial \Theta} \right] \tag{1}$$

where $w(t)$ is a weighting function, $\epsilon_\phi(\cdot)$ is the predicted noise by the 2D diffusion prior $\phi$, and $\Delta p$ is the relative camera pose change from the reference camera $r$. Additionally, we optimize the reference view image $I^r_{\text{RGB}}$ and transparency $I^r_{\text{A}}$ to align with the input:

$$\mathcal{L}_{\text{Ref}} = \lambda_{\text{RGB}} ||I^r_{\text{RGB}} - \tilde{I}^r_{\text{RGB}}||^2_2 + \lambda_{\text{A}} ||I^r_{\text{A}} - \tilde{I}^r_{\text{A}}||^2_2 \tag{2}$$

where $\lambda_{\text{RGB}}$ and $\lambda_{\text{A}}$ are the weights which are linearly increased during training. The final loss is the weighted sum of the above three losses.

**Text-to-3D.** The input for text-to-3D is a single text prompt. Following previous works, Stable-diffusion (Rombach et al., 2022) is used for the text-to-3D task. The SDS loss can be formulated as:

$$\nabla_\Theta \mathcal{L}_{\text{SDS}} = \mathbb{E}_{t,p,\epsilon} \left[ w(t)(\epsilon_\phi(I^p_{\text{RGB}}; t, e) - \epsilon) \frac{\partial I^p_{\text{RGB}}}{\partial \Theta} \right] \tag{3}$$

where $e$ is the CLIP embeddings of the input text description.

**Discussion.** We observe that *the generated Gaussians often look blurry and lack details even with longer SDS training iterations.* This could be explained by the ambiguity of SDS loss. Since each optimization step may provide inconsistent 3D guidance, it's hard for the algorithm to correctly densify the under-reconstruction regions or prune over-reconstruction regions as in reconstruction. This observation leads us to the following mesh extraction and texture refinement designs.

## 3.2 EFFICIENT MESH EXTRACTION

Polygonal mesh is a widely used 3D representation, particularly in industrial applications. Many previous works (Poole et al., 2022; Lin et al., 2023; Tsalicoglou et al., 2023; Tang et al., 2023a) export the NeRF representation into a mesh-based representation for high-resolution fine-tuning. We also seek to convert the generated 3D Gaussians into meshes and further refine the texture.

To the best of our knowledge, the polygonal mesh extraction from 3D Gaussians is still an unexplored problem. *Since the spatial density is described by a large number of 3D Gaussians, brute-force querying of a dense 3D density grid can be slow and inefficient.* It's also unclear how to extract the appearance in 3D, as the color blending is only defined with projected 2D Gaussians (Kerbl et al., 2023). Here, we propose an efficient algorithm to extract a textured mesh based on block-wise local density query and back-projected color.

**Local Density Query.** To extract the mesh geometry, a dense density grid is needed to apply the Marching Cubes (Lorensen & Cline, 1998) algorithm. An important feature of the Gaussian splatting algorithm is that over-sized Gaussians will be split or pruned during optimization. This is the foundation of the tile-based culling technique for efficient rasterization (Kerbl et al., 2023). We also leverage this feature to perform block-wise density queries.

We first divide the 3D space of $(-1, 1)^3$ into $16^3$ overlapping blocks, then cull the Gaussians whose centers are located outside each local block. This effectively reduces the total number of Gaussians to query in each block. We then query a $8^3$ dense grid inside each block, which leads to a final $128^3$ dense grid. For each query at grid position $\mathbf{x}$, we sum up the weighted opacity of each remained 3D Gaussian:

$$d(\mathbf{x}) = \sum_i \alpha_i \exp\left(-\frac{1}{2}(\mathbf{x} - \mathbf{x_i})^T \Sigma_i^{-1} (\mathbf{x} - \mathbf{x_i})\right) \tag{4}$$

where $\Sigma_i$ is the covariance matrix built from scaling $\mathbf{s}_i$ and rotation $\mathbf{q}_i$. An empirical threshold is then used to extract the mesh surface through Marching Cubes. Decimation and remeshing (Cignoni et al., 2008) are applied to post-process the extracted mesh to make it smoother and more compact.

**Color Back-projection.** Since we have acquired the mesh geometry, we can back-project the rendered RGB image to the mesh surface and bake it as the texture. We first unwrap the mesh's UV coordinates (Young, 2021) (detailed in Section A.1) and initialize an empty texture image. Then, we uniformly choose 8 azimuths and 3 elevations, plus the top and bottom views to render the corresponding RGB image. Each pixel from these RGB images can be back-projected to the texture

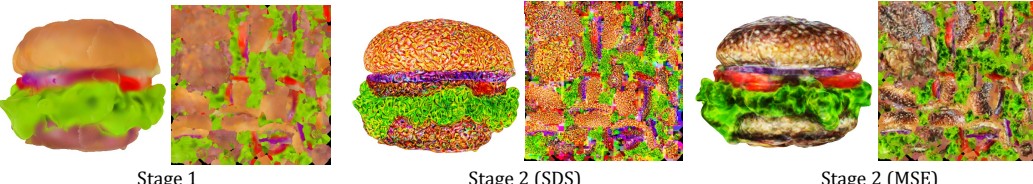

Stage 1                    Stage 2 (SDS)                    Stage 2 (MSE)

Figure 3: **Different Texture Fine-tuning Objectives**. We show that SDS loss produces artifacts for UV space texture optimization, while the proposed MSE loss avoids this.

image based on its UV coordinate. Following Richardson et al. (2023), we exclude the pixels with a small camera space z-direction normal to avoid unstable projection at mesh boundaries. This back-projected texture image serves as an initialization for the next texture fine-tuning stage.

### 3.3 UV-SPACE TEXTURE REFINEMENT

We further use a second stage to refine the extracted coarse texture. Different from texture generation (Richardson et al., 2023; Chen et al., 2023a; Cao et al., 2023), we hope to enhance the details given a coarse texture. However, fine-tuning the UV-space directly with SDS loss leads to artifacts as shown in Figure 3, which is also observed in previous works (Liao et al., 2023). This is due to the mipmap texture sampling technique used in differentiable rasterization (Laine et al., 2020). With ambiguous guidance like SDS, the gradient propagated to each mipmap level results in over-saturated color blocks. Therefore, we seek more definite guidance to fine-tune a blurry texture.

We draw inspiration from the image-to-image synthesis of SDEdit (Meng et al., 2021) and the reconstruction settings. Since we already have an initialization texture, we can render a blurry image $I_{\text{coarse}}^p$ from an arbitrary camera view $p$. Then, we perturb the image with random noise and apply a multi-step denoising process $f_\phi(\cdot)$ using the 2D diffusion prior to obtaining a refined image:

$$I_{\text{fine}}^p = f_\phi(I_{\text{coarse}}^p + \epsilon(t_{\text{start}}); t_{\text{start}}, c) \tag{5}$$

where $\epsilon(t_{\text{start}})$ is a random noise at timestep $t_{\text{start}}$, $c$ is $\Delta p$ for image-to-3D and $e$ for text-to-3D respectively. The starting timestep $t_{\text{start}}$ is carefully chosen to limit the noise strength, so the refined image can enhance details without breaking the original content. This refined image is then used to optimize the texture through a pixel-wise MSE loss:

$$\mathcal{L}_{\text{MSE}} = ||I_{\text{fine}}^p - I_{\text{coarse}}^p||_2^2 \tag{6}$$

For image-to-3D tasks, we still apply the reference view RGBA loss in Equation 2. We find that only about 50 steps can lead to good details for most cases, while more iterations can further enhance the details of the texture.

## 4 EXPERIMENTS

### 4.1 IMPLEMENTATION DETAILS

We train 500 steps for the first stage and 50 steps for the second stage. The 3D Gaussians are initialized to 0.1 opacity and grey color inside a sphere of radius 0.5. The rendering resolution is increased from 64 to 512 for Gaussian splatting, and randomly sampled from 128 to 1024 for mesh. The loss weights for RGB and transparency are linearly increased from 0 to $10^4$ and $10^3$ during training. We sample random camera poses at a fixed radius of 2 for image-to-3D and 2.5 for text-to-3D, y-axis FOV of 49 degree, with the azimuth in $[-180, 180]$ degree and elevation in $[-30, 30]$ degree. The background is rendered randomly as white or black for Gaussian splatting. For image-to-3D task, the two stages each take around 1 minute. We preprocess the input image by background removal (Qin et al., 2020) and recentering of the foreground object. The 3D Gaussians are initialized with 5000 random particles and densified for each 100 steps. For text-to-3D task, due to the larger resolution of $512 \times 512$ used by Stable Diffusion (Rombach et al., 2022) model, each stage takes around 2 minutes to finish. We initialize the 3D Gaussians with 1000 random particles and densify them for each 50 steps. For mesh extraction, we use an empirical threshold of 1 for

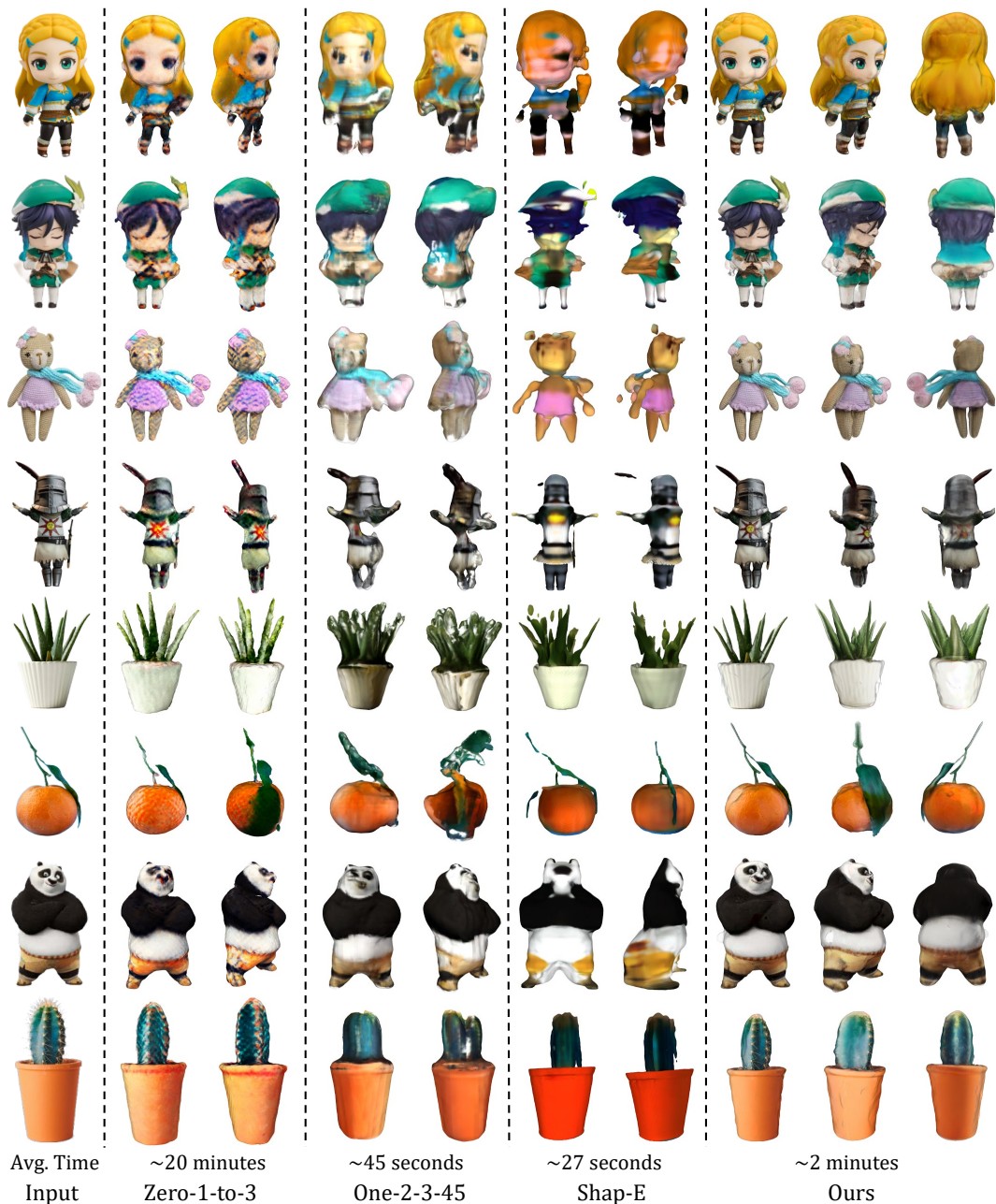

| Avg. Time | ~20 minutes | ~45 seconds | ~27 seconds | ~2 minutes |
| Input | Zero-1-to-3 | One-2-3-45 | Shap-E | Ours |

Figure 4: **Comparisons on Image-to-3D**. Our method achieves a better balance between generation speed and mesh quality on various images.

Marching Cubes. All experiments are performed and measured with an NVIDIA V100 (16GB) GPU, while our method requires less than 8 GB GPU memory. Please check the supplementary materials for more details.

## 4.2 QUALITATIVE COMPARISONS

We first provide qualitative comparisons on image-to-3D in Figure 4. We primarily compare with three baselines from both optimization-based methods (Liu et al., 2023b) and inference-only methods (Liu et al., 2023a; Jun & Nichol, 2023). For all compared methods, we export the generated models as polygonal meshes with vertex color or texture images, and render them under ambient

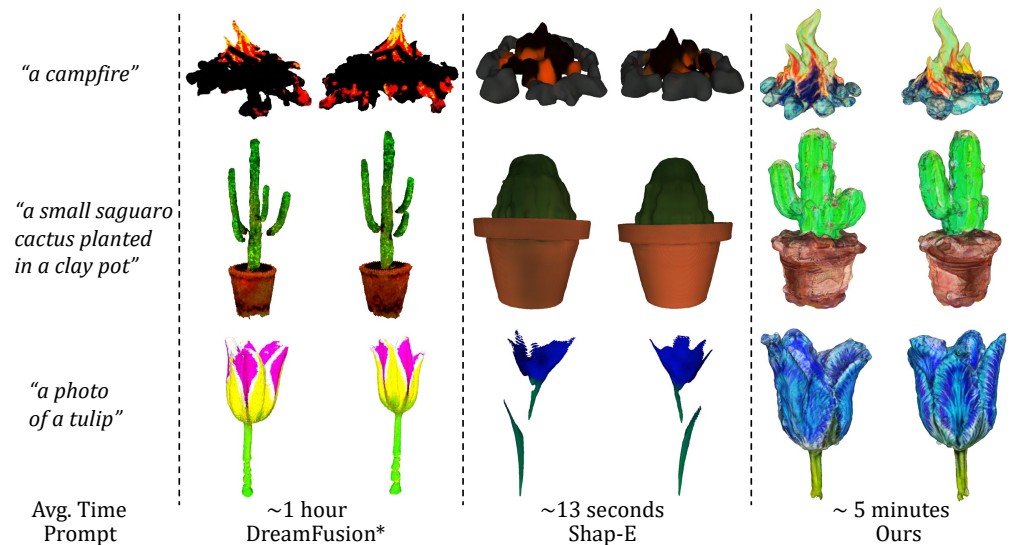

Figure 5: **Comparisons on Text-to-3D**. For Dreamfusion, we use the implementation from Guo et al. (2023) which also uses Stable-Diffusion as the 2D prior.

| | Type | CLIP-Similarity ↑ | Generation Time ↓ |
|---|---|---|---|
| One-2-3-45 (Liu et al., 2023a) | Inference-only | 0.594 | 45 seconds |
| Point-E (Nichol et al., 2022) | Inference-only | 0.587 | 78 seconds |
| Shap-E (Jun & Nichol, 2023) | Inference-only | 0.591 | 27 seconds |
| Zero-1-to-3 (Liu et al., 2023b) | Optimization-based | 0.647 | 20 minutes |
| Zero-1-to-3* (Liu et al., 2023b) | Optimization-based | 0.778 | 30 minutes |
| Ours (Stage 1 Only) | Optimization-based | 0.678 | 1 minute |
| Ours | Optimization-based | 0.738 | 2 minutes |

Table 1: **Quantitative Comparisons** on generation quality and speed for image-to-3D tasks. For Zero-1-to-3*, a mesh fine-tuning stage is used to further improve quality (Tang, 2022).

lighting. In terms of generation speed, our approach exhibits a noteworthy acceleration compared to other optimization-based methods. Regarding the quality of generated models, our method outperforms inference-only methods especially with respect to the fidelity of 3D geometry and visual appearance. In general, our method achieves a better balance between generation quality and speed, reaching comparable quality as optimization-based methods while only marginally slower than inference-only methods. In Figure 5, we compare the results on text-to-3D. Consistent with our findings in image-to-3D tasks, our method achieves better quality than inference-based methods and faster speed than other optimization-based methods. Furthermore, we highlight the quality of our exported meshes in Figure 6. These meshes exhibit uniform triangulation, smooth surface normals, and clear texture images, rendering them well-suited for seamless integration into downstream applications. For instance, leveraging software such as Blender (Community, 2018), we can readily employ these meshes for rigging and animation purposes.

### 4.3 QUANTITATIVE COMPARISONS

In Table 1, we report the CLIP-similarity (Radford et al., 2021; Qian et al., 2023; Liu et al., 2023a) and average generation time of different image-to-3D methods on a collection of images from previous works (Melas-Kyriazi et al., 2023; Liu et al., 2023a; Tang et al., 2023b) and Internet. We also conduct an user study on the generation quality detailed in Table 2. This study centers on the assessment of reference view consistency and overall generation quality, which are two critical aspects in the context of image-to-3D tasks. Our two-stage results achieve better view consistency and gener-

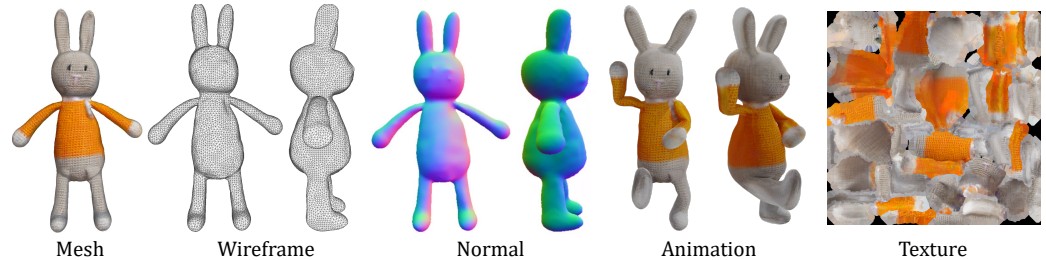

Mesh      Wireframe      Normal      Animation      Texture

Figure 6: **Mesh Exportation**. We export high quality textured mesh from 3D Gaussians, which can be seamlessly used in downstream applications like rigged animation.

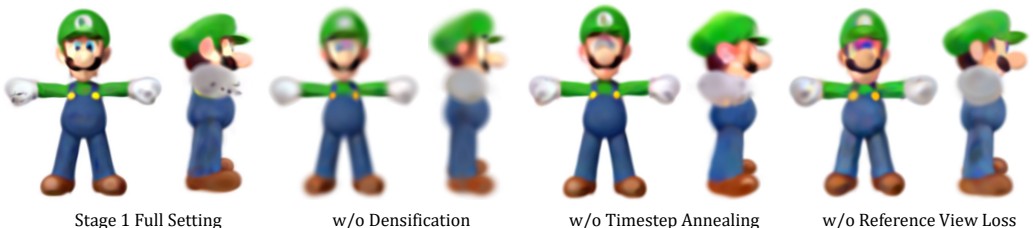

Stage 1 Full Setting      w/o Densification      w/o Timestep Annealing      w/o Reference View Loss

Figure 7: **Ablation Study**. We ablate the design choices in stage 1 training.

| | Zero-1-to-3 | One-2-3-45 | Shap-E | Ours |
|---|---|---|---|---|
| Ref. View Consistency ↑ | 3.48 | 2.34 | 1.80 | 4.31 |
| Overall Model Quality ↑ | 3.11 | 1.91 | 1.57 | 3.92 |

Table 2: **User Study** on image-to-3D tasks. The rating is of scale 1-5, the higher the better.

ation quality compared to inference-only methods. Although our mesh quality falls slightly behind that of other optimization-based methods, we reach a significant acceleration of over 10 times.

## 4.4 ABLATION STUDY

We carry out ablation studies on the design of our methods in Figure 7. We are mainly interested in the generative Gaussian splatting training, given that mesh fine-tuning has been well explored in previous methods (Tang et al., 2023a; Lin et al., 2023). Specifically, we perform ablation on three aspects of our method: **1)** Periodical densification of 3D Gaussians. **2)** Linear annealing of timestep $t$ for SDS loss. **3)** Effect of the reference view loss $\mathcal{L}_{\text{Ref}}$. Our findings reveal that omission of any design elements results in a degradation of the generated model quality. Specifically, the final Gaussians exhibit increased blurriness and inaccuracies, which further affects the second fine-tuning stage.

## 5 LIMITATIONS AND CONCLUSION

In this work, we present DreamGausssion, a 3D content generation framework that significantly improves the efficiency of 3D content creation. We design an efficient generative Gaussian splatting pipeline, and propose a mesh extraction algorithm from Gaussians. With our texture fine-tuning stage, we can produce ready-to-use 3D assets with high-quality polygonal meshes from either a single image or text description within a few minutes.

**Limitations.** We share common problems with previous works: Multi-face Janus problem, over-saturated texture, and baked lighting. It's promising to address these problems with recent advances in score debiasing (Armandpour et al., 2023; Hong et al., 2023), camera-conditioned 2D diffusion models (Shi et al., 2023; Liu et al., 2023c; Zhao et al., 2023; Li et al., 2023b), and BRDF auto-encoder (Xu et al., 2023b). Besides, the back-view texture generated in our image-to-3D results may look blurry, which can be alleviated with longer stage 2 training.

## ETHICS STATEMENT

We share common ethical concerns to other 3D generative models. Our optimization-based 2D lifting approach relies on 2D diffusion prior models (Liu et al., 2023b; Rombach et al., 2022), which may introduce unintended biases due to training data. Additionally, our method enhances the automation of 3D asset creation, potentially impacting 3D creative professionals, yet it also enhances workflow efficiency and widens access to 3D creative work.

## ACKNOWLEDGMENTS

This work is supported by the Sichuan Science and Technology Program (2023YFSY0008), National Natural Science Foundation of China (61632003, 61375022, 61403005), Grant SCITLAB-20017 of Intelligent Terminal Key Laboratory of SiChuan Province, Beijing Advanced Innovation Center for Intelligent Robots and Systems (2018IRS11), and PEK-SenseTime Joint Laboratory of Machine Vision. This study is also supported by the Ministry of Education, Singapore, under its MOE AcRF Tier 2 (MOE-T2EP20221- 0012), NTU NAP, and under the RIE2020 Industry Alignment Fund – Industry Collaboration Projects (IAF-ICP) Funding Initiative, as well as cash and in-kind contribution from the industry partner(s).

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

# A APPENDIX

## A.1 PRELIMINARY

**Score Distillation Sampling (SDS)**. SDS was initially introduced by Dreamfusion (Poole et al., 2022), providing a framework that leverages pretrained 2D diffusion models as priors to optimize a parametric image generator. A representative example involves employing a differentiable 3D representation, such as NeRF (Mildenhall et al., 2020), as the image generator:

$$\mathbf{x} = g_\Theta(p) \tag{7}$$

where $\mathbf{x}$ represents the rendered 2D image from the camera pose $p$, and $g_\Theta(\cdot)$ denotes the differentiable rendering function with optimizable NeRF parameters $\Theta$. The SDS formulation is expressed as:

$$\nabla_\Theta \mathcal{L}_{\text{SDS}} = \mathbb{E}_{t,p,\epsilon} \left[ w(t)(\epsilon_\phi(\mathbf{x}; t, e) - \epsilon) \frac{\partial \mathbf{x}}{\partial \Theta} \right] \tag{8}$$

where $t \sim \mathcal{U}(0.02, 0.98)$ is a randomly sampled timestep, $p$ is a randomly sampled camera pose orbiting the object center, $\epsilon \sim \mathcal{N}(0, 1)$ is a random Gaussian noise, $w(t) = \sigma_t^2$ is a weighting function from DDPM (Ho et al., 2020), $\epsilon_\phi(\cdot)$ is the noise predicting function with a pretrained parameters $\phi$, and $e$ is the text embedding. By optimizing this objective, the denoising gradient $(\epsilon_\phi(\mathbf{x}; t, e) - \epsilon)$ that contains the guidance information is back-propagated to the rendered image $\mathbf{x}$, which will be further back-propagated to the underlying NeRF parameters $\Theta$ through differentiable rendering (Mildenhall et al., 2020). Therefore, the NeRF can be optimized to form a 3D shape corresponding to the text description.

**UV Mapping**. UV Mapping is used to project a 2D texture image onto the surface of a 3D polygonal mesh. This requires to map each mesh vertex to a position on the image plane, which is stored as the UV coordinates for each vertex. UV unwrapping (Young, 2021) is employed to automatically compute these UV coordinates given a mesh. Retrieving the texture value at any surface point on a triangle involves barycentric interpolation to calculate the UV coordinate. We utilize NVdiffrast (Laine et al., 2020) for texture mapping and differentiable rendering, facilitating the optimization of the texture image through rendered images.

## A.2 MORE IMPLEMENTATION DETAILS

**Learning Rate**. For the learning rate of Gaussian splatting, we set different values for different parameters. The learning rate for position is decayed from $1 \times 10^{-3}$ to $2 \times 10^{-5}$ in 500 steps, for feature is set to $0.01$, for opacity is $0.05$, for scaling and rotation is $5 \times 10^{-3}$. For mesh texture fine-tuning, the learning rate for texture image is set to $0.2$. We use the Adam (Kingma & Ba, 2014) optimizer for both stages.

**Densification and Pruning**. Following Kerbl et al. (2023), the densification in image-to-3D is applied for Gaussians with accumulated gradient larger than $0.5$ and max scaling smaller than $0.05$. In text-to-3D, we set the gradient threshold to $0.01$ to encourage densification. We also prune the Gaussians with an opacity less than $0.01$ or max scaling larger than $0.05$.

**Mesh Extraction**. After extracting the mesh using Marching Cubes (Lorensen & Cline, 1998), we apply isotropic remeshing and quadric edge collapse decimation (Cignoni et al., 2008) to control the mesh complexity. Specifically, we first remesh the mesh to an average edge length of $0.015$, and then decimate the number of faces to $10^5$.

**Evaluation Settings**. We adopt the CLIP-similarity metric (Melas-Kyriazi et al., 2023; Qian et al., 2023; Liu et al., 2023a) to evaluate the image-to-3D quality. A dataset of 30 images collected from previous works (Melas-Kyriazi et al., 2023; Liu et al., 2023a; Tang et al., 2023b; Liu et al., 2023c) and Internet covering various objects is used. We then render 8 views with uniformly sampled azimuth angles $[0, 45, 90, 135, 180, 225, 270, 315]$ and zero elevation angle. These rendered images are used to calculate the CLIP similarities with the reference view, and we average the different views for the final metric. We use the `laion/CLIP-ViT-bigG-14-laion2B-39B-b160k`[1] checkpoint to calculate CLIP similarity. For the user study, we render 360 degree rotating videos

---

[1] https://huggingface.co/laion/CLIP-ViT-bigG-14-laion2B-39B-b160k

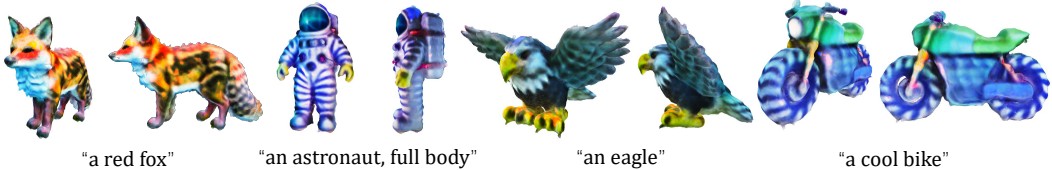

"a red fox"   "an astronaut, full body"   "an eagle"   "a cool bike"

Figure 8: **Text-to-3D results** with MVDream (Shi et al., 2023) as the guidance model.

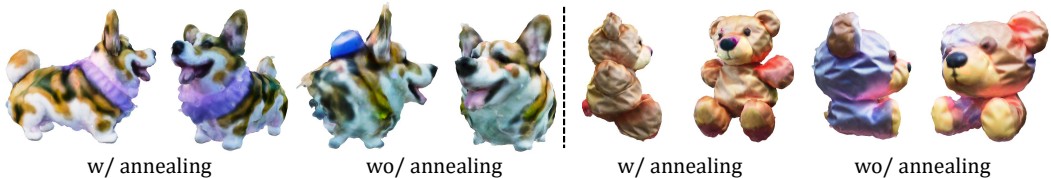

w/ annealing   wo/ annealing   w/ annealing   wo/ annealing

Figure 9: **Ablation on timestep annealing for text-to-3D**. We use MVDream (Shi et al., 2023) as the guidance model.

of 3D models generated from a collection of 15 images. There are in total 60 videos for 4 methods (Zero-1-to-3 (Liu et al., 2023b), One-2-3-45 Liu et al. (2023a), Shap-E Jun & Nichol (2023), and our method) to evaluate. Each volunteer is shown 15 samples containing the input image and a rendered video from a random method, and ask them to rate in two aspects: reference view consistency and overall model quality. We collect results from 60 volunteers and get 900 valid scores in total.

### A.3 MORE RESULTS

**Image-to-3D**. In Figure 10, we show more visualization results of our method. Specially, we compare the mesh output before and after our texture fine-tuning stage. We also compare against a SDS-based mesh fine-tuning method for Zero-1-to-3 (Liu et al., 2023b) noted as Zero-1-to-3* (Tang, 2022). Both stages of our method are faster than previous two-stage image-to-3D methods, while still reaching comparable generation quality. Our method also support images with non-zero elevations. As illustrated in Figure 11, our method can perform image-to-3D correctly with an extra estimated elevation angle as input. We make sure the random elevation sampling covers the input elevation and at least $[-30, 30]$ degree.

**Text-to-image-to-3D**. In Figure 13, we demostrate the text-to-image-to-3D pipeline (Liu et al., 2023a; Qian et al., 2023). We first apply text-to-image diffusion models (Rombach et al., 2022) to synthesize an image given a text prompt, then perform image-to-3D using our model. This usually gives better results compared to directly performing text-to-3D pipeline, and takes less time to generate. We show more animation results from our exported meshes in Figure 12.

**Text-to-3D with MVDream**. In Figure 8, we show text-to-3D results using the multi-view diffusion model MVDream (Shi et al., 2023) as the guidance. The multi-face Janus problem can be significantly mitigated by incorporating camera information to the 2D guidance model. However, it still suffers from over-saturation and unsmooth geometry. We further perform an ablation study on the linear timestep annealing in Figure 9. With the timestep annealing, we find the model converges to a more reasonable shape with the same amount of trianing iterations.

**Limitations**. We also illustrate the limitations of our method in Figure 14. Our image-to-3D pipeline may produce blurry back-view image and cannot generate fine details, which looks unmatched to the front reference view. With longer training of stage 2, the blurry problem of back view can be alleviated. For text-to-3D, we share common problems with previous methods, including the multi-face Janus problem and baked lighting in texture images.

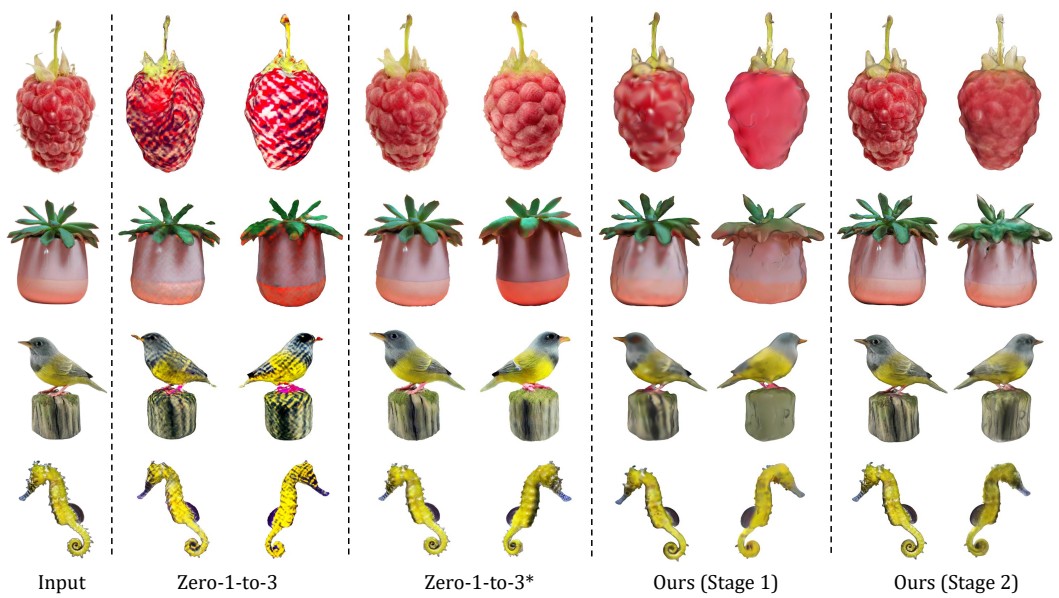

| Input | Zero-1-to-3 | Zero-1-to-3* | Ours (Stage 1) | Ours (Stage 2) |

Figure 10: **More Qualitative Comparisons**. We compare the results from two training stages of our method and Zero-1-to-3 (Liu et al., 2023b).

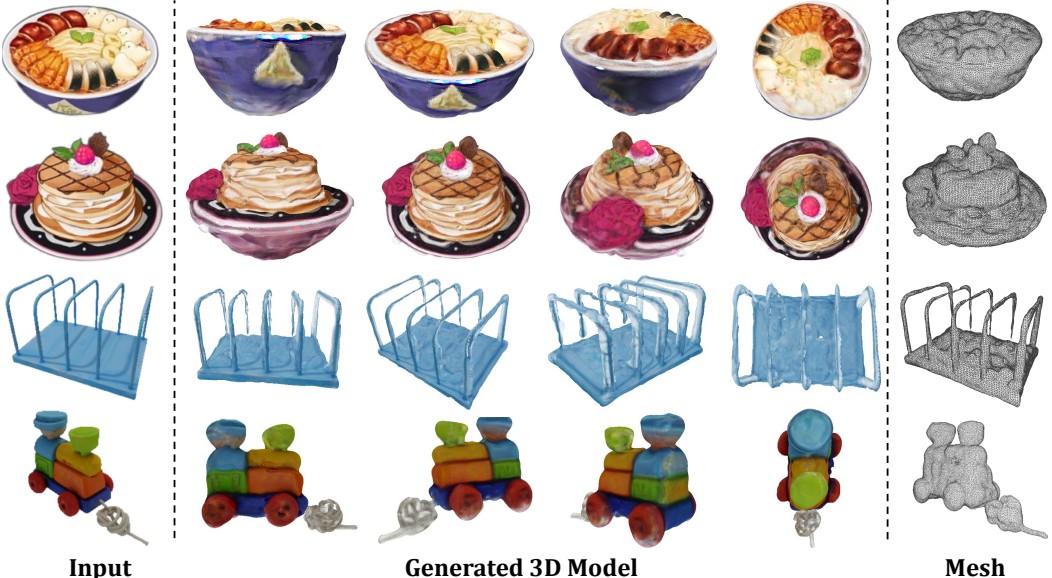

**Input**        **Generated 3D Model**        **Mesh**

Figure 11: **Results on images with different elevations**. Our method supports input images with a non-zero elevation angle.

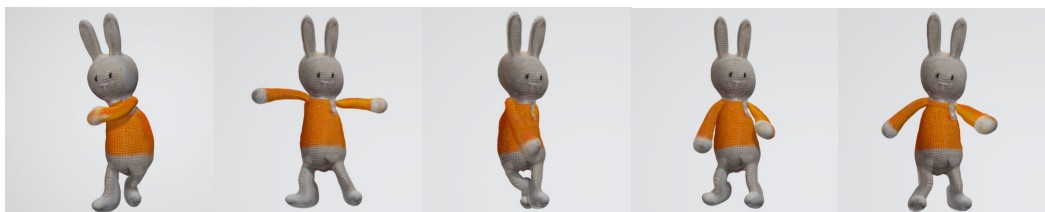

Figure 12: **Results on mesh animation**. Our exported meshes are ready-to-use for downstream applications like rigged animation.

| Prompt | Image | Generated 3D Model | | | Mesh |
|---|---|---|---|---|---|
| *"a nendoroid of a cute boy"* | | | | | |
| *"a nendoroid of a cute girl"* | | | | | |
| *"a penguin"* | | | | | |
| *"a potted cactus plant"* | | | | | |
| *"a 3D model of a fox"* | | | | | |
| *"a 3D model of a soldier"* | | | | | |

Figure 13: **Text-to-image-to-3D**. We first synthesize an image given a text prompt, then perform image-to-3D generation.

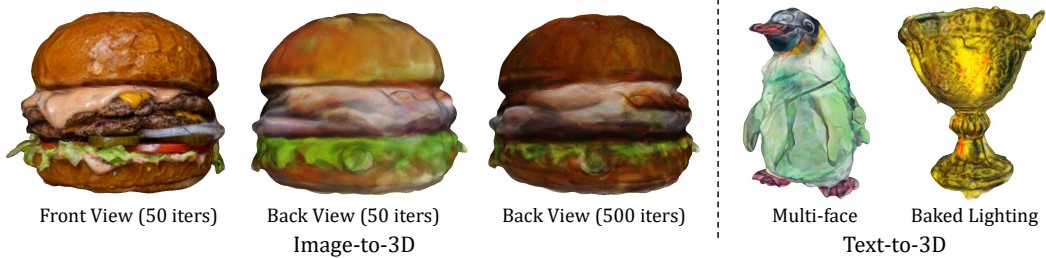

Figure 14: **Limitations**. Visualization of the limitations of our method.

