# OpenReview forum: "DreamGaussian: Generative Gaussian Splatting for Efficient 3D Content Creation"
_ICLR.cc/2024/Conference — ICLR 2024 oral_

### Official Review · Reviewer_MZY6 · 2023-10-31

**Soundness:** 3 good
**Presentation:** 2 fair
**Contribution:** 4 excellent
**Rating:** 8
**Confidence:** 4

**Summary:**

DreamGaussian is a novel 3D content generation framework designed to efficiently produce high-quality 3D content. The core innovation is a generative 3D Gaussian Splatting model paired with mesh extraction and texture refinement processes in UV space. Unlike the occupancy pruning seen in Neural Radiance Fields, this approach uses progressive densification of 3D Gaussians, which results in faster convergence for 3D generation tasks. The framework also incorporates an algorithm to transform 3D Gaussians into textured meshes and employs a fine-tuning stage for detail refinement. In tests, DreamGaussian was able to produce high-quality textured meshes from a single-view image in just 2 minutes, marking a tenfold speed improvement over existing methods.

**Strengths:**

The paper's main contribution is changing the NeRF representation to Gaussian Splatting representation in the current text-to-3D or image-to-3D pipeline. This has significant challenges in terms of implementation. Also, the authors' current implementation allows for fast generation of assets, which has significant importance in the 3D field.

**Weaknesses:**

Regarding the note “Similar to Dreamtime (Huang et al., 2023), we decrease the timestep t linearly”: Did the decrease of t help with the training time as well? Did it bring instability in training to the model? My previous exploration in this direction (for text to 3D in NeRF) showed improvement in the speed of generation for some objects but brought instability in the training. Since some objects needed more training steps than others, having a fixed annealing strategy damaged the performance, specifically for objects (e.g., motorcycles or dogs). It would be great if the authors could explore, in text to 3D, what the effect of the speed of annealing would be for different sets of prompts, (e.g., a corgi vs. a tree). Does the speed of annealing need to be tuned for each prompt?


The authors provided the experiment for time annealing in Fig 7 for image to 3D, but the paper is missing the same figure for text to 3D. Also, the effect of the speed of annealing needs to be explored.

Regarding the loss function, eq. 6, for UV-Space texture refinement: what happens if the object boundary in the Img2Img stage changes? How do the authors prevent the color of the background from leaking into the mesh color?

It would be great if the authors could compare the proposed texture refinement to other SOTA methods like TexFusion [1] or Text2Tex [2] or any other method of their choice. The reason for this ask is because texture refinement has been applied on top of the mesh. So, if the authors want to consider texture refinement as one of their contributions, they should either compare it with other baselines or reframe the paper and consider this as a side contribution.


I also tried experimenting with the code (great codebase!) and observed that, for text to 3D, the texture in most cases was very saturated. It would be great if the authors could comment on this phenomenon in the paper as a shortcoming and provide some insight into which parameter tuning might help.


Regarding the Janus problem, the authors provided a list of papers that address the Janus problem mostly using 3D data. However, it would be great if the authors could also reference methods like Perp-Neg [3] or Prompt-Debiasing [4] that address the Janus problem without the need for 3D assets. This has significant importance because they don't introduce bias from 3D data into the pipeline. For instance, they allow for the generation of a dog without the strict square position for standing that comes from the 3D asset.
It would be great if the authors could also comment on the guidance of the SDS loss and the effect of increasing those values.

There are many typos in the paper, and it would be great if the authors could fix them. Examples are: on page 2, "Gaussian splitting" appears to be a typo; it should likely be "Gaussian splatting". “severl methods” on page 3 and “Dissusion” on page 5 is it supposed to be discussion?

It would also be great if the authors could comment on how to get the model to consider both the input image and the text prompt simultaneously in both the initial stage and the later mesh/texture optimization stage. At the moment, it seems to ignore the optional text prompt. Was this intentional?

The paper contrary to DreamFusion does not learn an MLP for the background, it would be great if authors could comment on why they made that choice and what are their findings.

[1] https://openaccess.thecvf.com//content/ICCV2023/papers/Cao_TexFusion_Synthesizing_3D_Textures_with_Text-Guided_Image_Diffusion_Models_ICCV_2023_paper.pdf

[2]https://openaccess.thecvf.com/content/ICCV2023/papers/Chen_Text2Tex_Text-driven_Texture_Synthesis_via_Diffusion_Models_ICCV_2023_paper.pdf

[3] https://arxiv.org/abs/2304.04968

[4] https://arxiv.org/abs/2303.15413

**Questions:**

Please consider the comments in the weaknesses section. I believe the current paper as it presents a great contribution to the field, and by addressing my current comment I am willing to increase my score even more.

---

> ### Author Response · Authors · 2023-11-14
> **Reply to Reviewer MZY6**
>
> Thank you for your valuable time and insightful comments! We have tried to address your concerns in the updated manuscript and our rebuttal text:
>
> **Q1: Does the timestep annealing help in text-to-3D too?**
>
> Thanks for the advice! Since image-to-3D has a strong reference view image prior, we don't observe obvious instability during training.
>
> For text-to-3D, we have updated new text-to-3D results using MVDream [1] as the guidance model in the appendix. An ablation study shows that timestep annealing is still helpful in generating a more reasonable shape with the same amount of training iterations. However, we do observe that different input prompts may require different number of training iterations to converge, so a linear annealing may not be the optimal way.
>
>
>
> **Q2: In the refinement stage, what happens if the image-to-image diffusion changes the object boundary? How to prevent the color from background leaking into mesh?**
>
> Since we use a relatively small noise level for image-to-image diffusion, boundary change is not very obvious. Also, since the optimization is performed from multiple camera views, it could be corrected from another view should any color leaking happened in one view.
>
>
>
> **Q3: Comparison of the texture refinement stage with texture generation methods like Texfusion and Text2tex.**
>
> Thanks for reminding us! Although we both generate texture on a fixed mesh geometry, there are several differences:
>
> (1) These methods require a text prompt as the input. However, we don't have a prompt in our image-to-3D setting.
>
> (2) We have a coarse texture and we want to enhance the details based on it. These methods focus on texture generation instead of refinement and will ignore the coarse texture, which is not suitable especially for image-to-3D.
>
> We have updated the paper to discuss these methods.
>
>
>
> **Q4: Text-to-3D often generates over-saturated texture.**
>
> Thanks for mentioning this! We have updated the paper and stated this problem as a shortcoming in the limitations.
>
>
>
> **Q5: Missing references on methods addressing Janus problem without using 3D assets.**
>
> Thanks for reminding us! We have updated and discussed these methods in the paper.
>
>
>
> **Q6: There are many typos in the paper.**
>
> Thanks for correcting us! We have revised and updated the paper to fix these typos.
>
>
>
> **Q7: Why choose not to use a learnable background model?**
>
> The major reason is that we want to keep the pipeline simple, and we find that using random black or white background is enough for Gaussians to converge. We are not the first to choose this design, as Fantasia3D [2] also adopts a solid background color during optimization.
>
> The learnable background model is majorly adopted in NeRF-based method. One possible explanation is that NeRF tends to form the background during early optimization, which is undesired since we want NeRF to form the target object. However, we observe that it's less likely for mesh or Gaussian-based method with explicit geometry to form the background, so a learnable background model is unnecessary.
>
>
>
> [1] Shi, Yichun, et al. "Mvdream: Multi-view diffusion for 3d generation." *arXiv preprint arXiv:2308.16512* (2023).
>
> [2] Chen, Rui, et al. "Fantasia3d: Disentangling geometry and appearance for high-quality text-to-3d content creation." *arXiv preprint arXiv:2303.13873* (2023).
>
>
>
> We hope our responses satisfactorily address your queries. Please let us know to address any further concerns impacting your review.

---

> > ### Author Response · Authors · 2023-11-22
> > **Reply to Reviewer MZY6**
> >
> > We sincerely appreciate your great efforts in reviewing this paper. Your constructive advice and valuable comments really help improve our paper. Considering the approaching deadline, please, let us know if you have follow-up concerns. We sincerely hope you can consider our reply in your assessment, and we can further address unclear explanations and remaining concerns if any.
> >
> > Once more, we are appreciated for the time and effort you've dedicated to our paper.

---

> ### Comment · Reviewer_MZY6 · 2023-12-03
> **After rebuttel**
>
> Dear Authors,
>
> Thank you for addressing my previous feedback in your rebuttal.
>
> I acknowledge the emphasis your paper places on the speed of generation, which is indeed a significant contribution. However, I must express my reservations regarding the quality of the output. Compared to other concurrent works, the quality here seems to be notably inferior. This aspect is particularly critical, as it undermines the overall impact of the work.
>
> Furthermore, while the contribution towards texture enhancement is noted, its value is somewhat diminished due to the overall low quality of the output. This limitation prevents the texture contribution from being as meaningful as it could be.
>
> In light of these observations, I have decided to maintain my original score for the paper. **I believe the paper should be accepted due to its notable contribution in terms of generation speed.** However, given the concerns regarding output quality and the relatively minor impact of the texture contribution, **I do NOT recommend it for selection as one of the top papers at ICLR.**
>
> I hope this feedback is helpful for your future endeavors in research.
>
> Best regards,

---

### Official Review · Reviewer_huut · 2023-10-31

**Soundness:** 3 good
**Presentation:** 3 good
**Contribution:** 3 good
**Rating:** 8
**Confidence:** 3

**Summary:**

This paper proopsed a way to combine SDS loss and the recently proposed point-based rendering method GS. The pipeline is composed of 3 stages, 1) gaussian splatting optimization; 2) mesh extraction from point clouds; 3) texture refinement. The first stage is similar to other SDS-based methods which require pretrained 2d image diffusion models. However, the rendering method is switched to gaussian splatting. The second stage is done by applying marching cubes to opacities learned in the first stage. In the last stage, the texture is refined using 2D diffusion models. The full pipeline takes several minutes and we can obtain a mesh with textures. The authors show some results of image-conditioned and text-conditioned generation.

**Strengths:**

The paper has many strengths. Thanks to GS, the full pipeline is fast compared to some previous nerf-based methods. The author observed that a longer optimization of SDS does not give better results (sharp and detailed). Thus the focus of the method is not the SDS part. Instead, the authors only optimize the SDS loss with a few iterations (which is also the main reason why it is so fast). After obtaining the blurry 3D object, a refinement step inspired by diffusion-based image editing is applied. In the end, we can have a detailed mesh with textures. Another important aspect of this method, is the mesh extraction algorithm. Extracting a surface from a point cloud is not straightforward. The authors found out a way to use the opacity as the isosurface.

When we are generating data, GS seems to be more suitable because of its progressive nature. The paper can be seen as a proof of this claim.

**Weaknesses:**

1. It seems the focus of the paper is image-conditioned generation. The results of text-to-3d are very limited and the comparison is weak.
2. I am curious about the setup of the stage 3. If we already have a mesh with coarse texture, we can optimize it with differentiable mesh rendering and SDS loss, as Fantasia3d did in the appearance modeling stage. What is the advantage of the proposed refinement compared to this?
3. Another concurrent ICLR submission ( https://openreview.net/forum?id=pnwh3JspxT ) optimizes SDS much longer (1 hour and 40 minutes) than this paper. However, this paper claims that longer training does not give better results. Can the authors clarify the differences?

**Questions:**

See weakness section.

---

> ### Author Response · Authors · 2023-11-14
> **Reply to Reviewer huut**
>
> Thank you for your valuable time and insightful comments! We have tried to address your concerns in the updated manuscript and our rebuttal text:
>
> **Q1:  The text-to-3D results are limited.**
>
> Thanks for your advice! Our paper does focus more on image-to-3D, since text-to-3D generation takes longer computing time and is more prone to suffer from the Jasus problem.
>
> As stated in the limitations, it's promising to solve the Janus problem with recent multi-view or camera-conditioned 2D diffusion models. We have updated new text-to-3D results using MVDream [1] as the guidance model in the appendix, which shows better results for difficult prompts.
>
>
>
> **Q2: What is the advantage of the proposed refinement stage compared to Fantasia3D?**
>
> The key advantage of our refinement stage lies in its efficiency.
>
> Unlike other methods, which typically start without initial textures and require considerable time for generation, our approach builds on an existing mesh with coarse textures. For example, the texturing stage in Fantasia3D can take up to 20 minutes using 8 GPUs, whereas our refinement stage enhances the coarse texture in just 1 minute on a single GPU.
>
>
>
> **Q3: Differences compared to GSGEN which optimizes for longer time.**
>
> There are several different designs that lead to the different conclusions about optimization time:
>
> (1) GSGEN prioritizes high-quality text-to-3D generation, which is more challenging than image-to-3D. To address the Janus problem, they also generate a point cloud using Point-E [2] to initialize the Gaussians.
>
> (2) They introduce a novel compactness-based densification strategy, which is more suitable for longer optimization.
>
> (3) In addition to the 2D SDS loss using Stable-diffusion, GSGEN applies a 3D SDS loss with Point-E [2], potentially prolonging each optimization step.
>
> In contrast, our method emphasizes efficiency, trading off some level of detail. We also observe that text-to-3D typically requires longer optimization to achieve finer details, especially with an appropriate densification strategy.
>
>
>
> [1] Shi, Yichun, et al. "Mvdream: Multi-view diffusion for 3d generation." *arXiv preprint arXiv:2308.16512* (2023).
>
> [2] Nichol, Alex, et al. "Point-e: A system for generating 3d point clouds from complex prompts." *arXiv preprint arXiv:2212.08751* (2022).
>
>
>
> We hope our responses satisfactorily address your queries. Please let us know to address any further concerns impacting your review.

---

> > ### Author Response · Authors · 2023-11-22
> > **Reply to Reviewer huut**
> >
> > We sincerely appreciate your great efforts in reviewing this paper. Your constructive advice and valuable comments really help improve our paper. Considering the approaching deadline, please, let us know if you have follow-up concerns. We sincerely hope you can consider our reply in your assessment, and we can further address unclear explanations and remaining concerns if any.
> >
> > Once more, we are appreciated for the time and effort you've dedicated to our paper.

---

### Official Review · Reviewer_E7Ru · 2023-11-01

**Soundness:** 3 good
**Presentation:** 4 excellent
**Contribution:** 4 excellent
**Rating:** 10
**Confidence:** 5

**Summary:**

The paper addresses the problem of 3D content generation from text- or single view inputs into 3D gaussian splats which can further be transformed into textured meshes. The proposed method is coined DreamGaussian and revisits the recent ToG 2023 3D Gaussian Splatting paradigm with a generative twist to it.

The proposed contribution is favorably compared to a comprehensive set of state-of-the-art comparative baselines and in particular is able to produced high fidelity mehses of objects within 2min of compute time, which is remarkable.

**Strengths:**

+ ## Readability.
As it currently stands, the paper is very well written. The main ideas and concepts are mostly well explained and articulated throuthout.

+ ## Organization of the contents and overall paper structure.
The contents are also very well structured and balanced.

+ ## Overall maturity of the submitted package, which makes it very reasonably within camera ready territory.

+ ## The actual performance of the proposed methodological contribution.
In particular, it dramatically cuts down the computation time in the space of optimization-based techniques in the field, by an order of magnitude.

+ ## Related work section and discussion.
It is very well structued, articulated and populated with very relevant and up to date references.

**Weaknesses:**

+  ## 1. Missing bits of context information - How much does it cost?
While indicative timings and implementation details (covering experimental setup,  are provided, information regarding the resource usage, model size and complexity are currently underdescribed.

A comparative disclosure of such information covering the main experimental baselines that are considered would help the reader better assess its relative positioning throughout the typical criteria.

Mentioning where the computation bottlenecks lie in terms of pipeline components would also be valuable in order to fully assess the practical usefullness of the proposed sequential pipeline, beyond rough timings.

+  ## 2. Challenging the ad hoc meshing post processing.

As it currently stands, the  mesh extraction technique relies on many subsequent post-processing steps, including mesh decimation and remeshing (end of page 5 in the main paper). I believe the explainations around eq. (4) (before and after) could be further improved and detailed. My current intuition is that the mesh complexity at least could be controled jointly during the density query step. Also, given the lack of statistics given regarding each step (Weakness 1 above), the relative need and ROI to fuse these steps is also hard to assess.

The current procedure also produces non-manifold and non-watertight meshes with arbitrary complexity.

+  ## 3. Evaluation.
The user study (ie, subjective quality analysis) that is presented in the main paper and further detailed in the appendix is a good idea and often overlooked in the field.

Nevertheless, its size and statistical informative validity are rather limited as they are based on a "cohort" of 40 users and 15 input samples to assess from.

**Questions:**

The main questions I would have cover the aforementioned weaknesses that have been pinpointed. In particular regarding the missing bits of informations.

Besides those remaining grey areas, I would be happy to bump my initial rating were they to be addressed accordingly.

---

> ### Author Response · Authors · 2023-11-14
> **Reply to Reviewer E7Ru**
>
> Thank you for your valuable time and insightful comments! We have tried to address your concerns in the updated manuscript and our rebuttal text:
>
> **Q1: Missing details about the model size, complexity, computation bottlenecks.**
>
> Thanks for reminding us!
>
> The size of the generated 3D models varies depending on the input and training schedule. Typically, image-to-3D results involve around $10^4$ 3D Gaussians, while text-to-3D can reach up to $10^5$ Gaussians due to a more aggressive densification strategy.
>
> For the mesh extraction, we apply remeshing and decimation as post-processing so the size is controlled. Specifically, we perform isotropic remeshing to an average edge length of $0.015$, and then decimate the number of faces to $10^5$.
>
> The primary computational bottleneck in our pipeline remains the forwarding of the 2D diffusion model. We have broken down the time consumption for each operation per iteration as follows:
>
> | Operation | Render 3D Gaussians | Run diffusion model | Loss backward |
> | --------- | ------------------- | ------------------- | ------------- |
> | Time (ms) | 3.8                 | 58.4                | 26.4          |
>
> Our method primarily reduces generation time by requiring fewer iterations to converge (from several thousand to 500 steps). However, each iteration is still constrained by the 2D diffusion model.
>
> For the mesh extraction, the time consumption for each operation is detailed as:
>
> | Operation | Extract geometry | Unwarp UV | Extract texture |
> | --------- | ---------------- | --------- | --------------- |
> | Time (s)  | 7.9              | 4.6       | 4.0             |
>
>
>
> **Q2: How to control the mesh complexity? How to solve non-watertight/manifold meshes?**
>
> As mentioned in response to Q1, we control mesh complexity through post-processing, limiting the number of faces to $10^5$ through decimation. These post-processing details have been added to the implementation details.
>
> We acknowledge the potential for non-manifold or non-watertight meshes in cases of poor Gaussian convergence, but we consider repairing it is less related to the goal of this paper and leave it for future work.
>
>
>
> **Q3: The size of user study is limited.**
>
> Thanks for the advice! We have conducted additional surveys with 20 more participants, bringing the total to 60 users. The updated results are included in the paper. Due to time constraints in the rebuttal period, we plan to release the data from our experiments for public comparison to facilitate broader assessment and validation.
>
> We hope our responses satisfactorily address your queries. Please let us know to address any further concerns impacting your review.

---

> > ### Author Response · Authors · 2023-11-22
> > **Reply to Reviewer E7Ru**
> >
> > We sincerely appreciate your great efforts in reviewing this paper. Your constructive advice and valuable comments really help improve our paper. Considering the approaching deadline, please, let us know if you have follow-up concerns. We sincerely hope you can consider our reply in your assessment, and we can further address unclear explanations and remaining concerns if any.
> >
> > Once more, we are appreciated for the time and effort you've dedicated to our paper.

---

### Official Review · Reviewer_L9mE · 2023-11-06

**Soundness:** 3 good
**Presentation:** 1 poor
**Contribution:** 2 fair
**Rating:** 8
**Confidence:** 4

**Summary:**

The present work presents a new methodology for generative 3d modelling using a 2d lifting approach. Its main underlying hypothesis is that using 3d Gaussian Splatting with its densification results in much faster convergence compared to using traditional neural radiance fields. On the technical side, this work focuses on two main contributions. First it proposes a mesh extraction technique based on the marching cubes algorithm for the setting of environment representations using 3d gaussians. Second, it proposes a UV-space texture refinement stage for further quality enhancement of the resulting textures.

**Strengths:**

Overall I think this paper to be interesting and to make relevant contributions. While most of the underlying ideas have already been presented in prior works, their combination for the presented use-case is relevant and results in significant performance improvement for the tasks of image-to-3d and text-to-3d.

**Weaknesses:**

While the work has overall a good contribution, one of my main concern is its writing. First, there are numerous minor language issues such as grammar mistakes and sometimes unnecessarily complicated sentence structure. This can easily be solved by a few rounds of careful proofreading. Second, the work reads more like a paper written for a computer vision conference without providing sufficient background to a broader audience at the ICLR community. While this style is not unprecedented in ML, it makes this work much harder accessible and misses an opportunity. Moreover, for people outside the specific subarea of 3d content creation some of the important implementation details may be missing.

These concerns range from minor points such as assuming the reader to be familiar with all mentioned vision / graphics concepts such as UV space etc without providing a proper background section. It also involves more complicated points such as the decision to provide some background (e.g. on SDS loss) but only to an extent that is only meaningful for people already familiar with dreamfusion. While in vision, many of these things can be assumed known, it would be useful to the learning community to provide some information here.

Also, I would rephrase the contribution bullet points to focus stronger on the technical aspects rather than first mentioning the overall framework, then dedicating one point to the actual technical meat and then talking about experimental evaluation.

**Questions:**

* Maybe rephrase the formulation "to release the potential of optimization-based methods." to something like "to unlock the potential. ..." or something similar.
* When writing "we decrease the timestep t linearly, which is used to weight the random noise ϵ added to the rendered RGB image", it is not fully clear to me how this is performed?
* The SDS loss formulation is not clear without knowing the sds loss, e.g. the expectation is taken among other variables over p and t. What is the distribution of p and t?
* Also, the evaluation section should be more explicit / more structured about the evaluation protocol and datasets used.

---

> ### Author Response · Authors · 2023-11-14
> **Reply to Reviewer L9mE**
>
> Thank you for your valuable time and insightful comments! We have tried to address your concerns in the updated manuscript and our rebuttal text:
>
> **Q1: The writing is poor and misses background details.**
>
> Thanks for your advice! The paper has been revised to enhance the writing quality. Additionally, we have included a preliminary section in the appendix to introduce relevant background details including SDS loss and mesh UV mapping.
>
>
>
> **Q2: What is the distribution of p and t in SDS loss?**
>
> $p$ represents the camera pose to render the 3D Gaussians, which is uniformly sampled to orbit the object center.
>
> For example, for image-to-3D, the camera is sampled with a radius of $2$, a y-axis FOV of $46$ degree, with the azimuth in $[-180, 180] $ degree and the elevation in $[-30, 30]$ degree.
>
> $t$ represents the timestep in denoising diffusion probabilistic model.
>
> Early works like Dreamfusion [1] samples $t$ from $[0.02, 0.98]$ uniformly during training. We follow Dreamtime [2] to perform an annealing of  $t$ from $0.98$ to $0.02$ during training, which leads to faster convergence as shown in the ablation study (Figure 7).
>
>
>
> **Q3: How to perform the linear timestep decreasing?**
>
> As answered in Q2, we linearly decrease $t$ from $0.98$ to $0.02$ with iteration step $i$ during training.
>
> This can be viewed a simplification of the TP-SDS algorithm proposed in Dreamtime [2]. Their findings suggest that larger values of $t$ are crucial for the global structure, whereas smaller values enhance local details. Hence, a decreasing schedule aligns the SDS noise level with NeRF optimization, facilitating a coarse-to-fine generation process.
>
>
>
> **Q4: More details about the evaluation protocol and datasets.**
>
> We have expanded on our evaluation methodology in section A.1 of the appendix.
>
> Given the absence of ground truth for 3D generative tasks, assessing generation quality poses a challenge. We follow previous works and use CLIP-similarity for evaluation. This metric calculates the average cosine similarity of CLIP embeddings between the input image and novel views from the generated 3D object. Our evaluation was conducted on a dataset comprising 30 samples.
>
>
>
> [1] Poole, Ben, et al. "Dreamfusion: Text-to-3d using 2d diffusion." *arXiv preprint arXiv:2209.14988* (2022).
>
> [2] Huang, Yukun, et al. "DreamTime: An Improved Optimization Strategy for Text-to-3D Content Creation." *arXiv preprint arXiv:2306.12422* (2023).
>
>
>
> We hope our responses satisfactorily address your queries. Please let us know to address any further concerns impacting your review.

---

> > ### Comment · Reviewer_L9mE · 2023-11-17
> >
> > > Thanks for your advice! The paper has been revised to enhance the writing quality. Additionally, we have included a preliminary section in the appendix to introduce relevant background details including SDS loss and mesh UV mapping.
> >
> > Thanks, the main text should probably refer somewhere to the new appendix content. Some thoughts on those:
> >
> > * In the explanation of the SDS loss, mayb mention how $\partial \mathbf{x} / \partial \Theta$ is computed in practice?
> >
> > * At the beginning of the UV Mapping section, I would probably write what it is used for, i.e. instead of starting by _" To project a 2D texture image onto the surface of a 3D mesh, it is essential to map each vertex to a position on the image plane"_ you could start by _"UV Mapping is used to project a 2d texture image onto the surface of a 3d mesh. This requires to map each vertex ..."_

---

> > > ### Author Response · Authors · 2023-11-18
> > > **Reply to Reviewer L9mE**
> > >
> > > Thank you for your valuable feedback! We have revised the paper according to your suggestions.
> > > Specifically, we have properly refered to the preliminary section in the main paper, detailed the gradient propagation to $\Theta$, and reformulated the introduction of the UV mapping section.

---

### Meta-Review · Area_Chair_ekVf · 2023-12-11

**Metareview:**

The paper has received unanimous recommendations for acceptance with ratings of 8/8/10/8. The rebuttal process has effectively addressed most concerns raised in the initial reviews. Reviewers collectively agree that the paper significantly reduces optimization time with a novel Gaussian Splatting scene rendering method in its pipeline. The Area Chair (AC) concurs with the assessment and recommends accepting the paper.

**Justification For Why Not Higher Score:**

N/A.

**Justification For Why Not Lower Score:**

The paper presents an efficient and complete 3D generation pipeline with a novel choice with Gaussian Splatting scene rendering method. It is a timely article for advancing the 3D generation field with effective new scene representation and rendering method as well as a complete pipeline for generating high resolution texture mesh for downstream 3D applications.

---

### Decision · Program_Chairs · 2024-01-16

Accept (oral)